# RAS-pathway mutation patterns define epigenetic subclasses in juvenile myelomonocytic leukemia

Daniel B. Lipka et al.[#]

Juvenile myelomonocytic leukemia (JMML) is an aggressive myeloproliferative disorder of early childhood characterized by mutations activating RAS signaling. Established clinical and genetic markers fail to fully recapitulate the clinical and biological heterogeneity of this disease. Here we report DNA methylome analysis and mutation profiling of 167 JMML samples. We identify three JMML subgroups with unique molecular and clinical characteristics. The high methylation group (HM) is characterized by somatic *PTPN11* mutations and poor clinical outcome. The low methylation group is enriched for somatic *NRAS* and *CBL* mutations, as well as for Noonan patients, and has a good prognosis. The intermediate methylation group (IM) shows enrichment for monosomy 7 and somatic *KRAS* mutations. Hypermethylation is associated with repressed chromatin, genes regulated by RAS signaling, frequent co-occurrence of RAS pathway mutations and upregulation of *DNMT1* and *DNMT3B*, suggesting a link between activation of the DNA methylation machinery and mutational patterns in JMML.

#A full list of authors and their affliations appears at the end of the paper

Juvenile myelomonocytic leukemia (JMML) is a myeloproliferative disorder (MPD) of early childhood that originates from the multipotent hematopoietic stem and progenitor cell (HSPC) compartment. JMML is characterized by overproduction of mature and immature myeloid cells, including the erythroid lineage[1]. Without adequate treatment, survival for most children is < 1 year. Although few cases show spontaneous remission, allogeneic hematopoietic stem cell transplantation (HSCT) remains the only curative treatment option for the majority of patients[2]. Yet, even with HSCT the 5-year event-free survival (EFS) still reaches only about 50%[3].

Hyperactive RAS signaling is assumed to be the main driving event in JMML. It is caused by somatic mutations in KRAS, NRAS, or PTPN11 in about 50% of patients[1,2,4–6]. In addition, 10–15% of JMML patients show clinical signs of neurofibromatosis with biallelic inactivation of the NF1 locus in leukemic cells[7], and another 10–15% have an underlying developmental disorder with germline CBL mutations (herein termed "CBL syndrome") and acquired loss of heterozygosity at the locus in leukemic cells[5,8]. Moreover, some patients with Noonan syndrome develop a self-limiting MPD, which is hematologically indistinguishable from JMML[9]. Recently, less frequent recurrent mutations in JAK3, RAC, and RRAS have been identified[10–12]. These new mutations also result in activation of intracellular signaling pathways including RAS and JAK/STAT signaling. RAS pathway mutations generally occur in a mutually exclusive manner in JMML and co-occurrence has only been described in some cases[10–12]. Together, the emerging picture of genetic alterations suggests underlying signaling defects involving the RAS pathway in almost all cases of JMML.

So far, there is no clear understanding of how RAS pathway mutations relate to the heterogeneous disease biology and variable clinical outcome seen in JMML patients. For example, some studies reported that JMML patients with NRAS mutations have a rather favorable course, including some cases with spontaneous disease regression[13,14]. In contrast, JMML with somatic PTPN11 mutations appear to represent cases with aggressive biology and are associated with a high risk of relapse after HSCT[15]. However, the affected RAS pathway gene alone does not fully explain clinical outcome. Evidence from the literature suggests that oncogenic RAS-signaling is able to modify epigenetic patterns[16–18]. Indeed, investigations of DNA methylation in JMML at the level of candidate gene promoters (AKAP12, BMP4, CALCA, CDKN2A, RARB, and RASA4) identified DNA hypermethylation to be associated with poor clinical outcome[19–21]. Still, to date, a comprehensive characterization of the DNA methylome in JMML is missing.

In the present study, we perform an integrative analysis of genome-wide DNA methylation profiles with mutational patterns, copy-number changes, and gene expression in primary JMML samples. This analysis uncovers distinct DNA methylation signatures, which are related to RAS pathway mutation patterns.

## Results

### Identification of JMML-specific methylation events.
Samples from 19 JMML patients and from 1 child with Noonan syndrome and MPD, together referred to as JMML samples (discovery cohort; Supplementary Table 1), were analyzed using the HumanMethylation450 Bead Chip Array (Illumina) to systematically investigate their DNA methylomes. Unsupervised hierarchical consensus clustering of the most variable CpG sites across all JMML samples revealed two clusters that separated the JMML samples (Supplementary Fig. 1a). In order to test the hypothesis that the two clusters reflect different cells-of-origin along the hematopoietic differentiation trajectory, the data were mapped in relation to methylomes of normal cell populations across all hematopoietic differentiation stages (Fig. 1a and Supplementary Fig. 1b–e). In this analysis, JMML samples formed a separate cluster, distinct from HSPCs, mature myeloid cells, B-cells and NK-cells, demonstrating that JMML methylomes reflect patterns both from terminally differentiated blood cells, as well as from more immature hematopoietic progenitor cells. Although the JMML methylomes exhibited more variability than the normal hematopoietic cell populations, the differentiation stage achieved did not disclose obvious JMML subgroups (Fig. 1a and Supplementary Fig. 1b–e). The observed elevated variability across the JMML methylomes could potentially be explained by differences in cell type composition across samples, which could mask JMML-specific DNA methylation patterns. Indeed, estimation of cell type contributions in individual JMML samples using reference methylomes from several normal hematopoietic cell types revealed substantial heterogeneity in cell composition across all JMML samples, with B-cells, HSPCs, and granulocytes showing the highest degree of variability (Fig. 1b)[22].

This observation prompted us to develop a strategy to identify JMML-specific methylation events that would not be affected by the samples' cell type composition. All CpGs showing dynamic methylation changes during normal hematopoietic differentiation, so called hematopoiesis-specific differentially methylated probes, were excluded from further analysis to retain only non-variable CpGs (nvCp2Gs; Fig. 1c and Supplementary Data 1). Consensus clustering of the discovery cohort using only nvCpGs identified two clusters that stably separated the JMML samples (Fig. 1d and Supplementary Fig. 1f–h). The first subgroup, named high methylation group (HM, n = 14 samples), showed significantly higher DNA methylation levels in the most variable nvCpGs as compared to the low methylation group (LM; Fig. 1e and Supplementary Fig. 1i). The methylation differences were most pronounced in CpG islands (Fig. 1e), suggesting a CpG-island methylator phenotype (CIMP), which has also been reported in other malignancies[23–25]. Supervised differential methylation analysis determined 5,380 JMML-specific differentially methylated probes (jmmlDMPs) separating the two JMML subgroups (Supplementary Data 2). The majority of jmmlDMPs (5,277; 98.1%) showed increased methylation in the HM group as compared with the LM group, whereas only 103 jmmlDMPs (1.9%) lost methylation. Hierarchical clustering of the jmmlDMPs faithfully recapitulated the identified JMML subgroups and showed a remarkable enrichment of relapses after HSCT (7/13 patients, 53.8%; one patient had missing information) in the HM group, while none of the patients in the LM group (0/6) had disease recurrence (Fig. 1f). Together, these data suggest that JMML-specific aberrant DNA methylation patterns might be associated with distinct clinical and biological features.

### DNA methylation correlates with clinical and genetic features.
To validate the methylation groups identified in the discovery cohort and to investigate their clinical relevance, an extended methylome analysis was performed in an unselected sample set consisting of 147 consecutive patients with JMML or Noonan syndrome and MPD registered in the EWOG-MDS 1998 or EWOG-MDS 2006 trials (validation cohort; Supplementary Table 2, Supplementary Data 3). Consensus clustering of the jmmlDMPs now identified, in addition to the HM and LM groups, a third subgroup of patients showing intermediate methylation levels (IM group; Supplementary Fig. 2a–d). Remarkably, the LM cluster was enriched for patients known to have low-risk disease[26], including all patients diagnosed with Noonan syndrome and MPD (18, 100%), all JMML patients

carrying *CBL* mutations (13) and the majority of patients with *NRAS* mutations (14/19, 73.7%; Fig. 2). Factors known to predict an unfavorable disease course were underrepresented in the LM group, which was reflected by a lower age at diagnosis (median age: 0.4 years; age > 2 years: 6%), lower rates of thrombocytopenia (platelets < 70 K μl$^{-1}$: 29%) and fewer patients with elevated

levels of fetal hemoglobin (proportion of cases with elevated HbF: 29%). By contrast, the HM cluster was enriched for patients showing high-risk characteristics: 70% carried somatic *PTPN11* mutations, 78% had low platelet counts (< 70 K μl$^{-1}$), all cases informative for HbF had elevated levels when adjusted for age and 88% were older than 2 years at the time of diagnosis. The IM

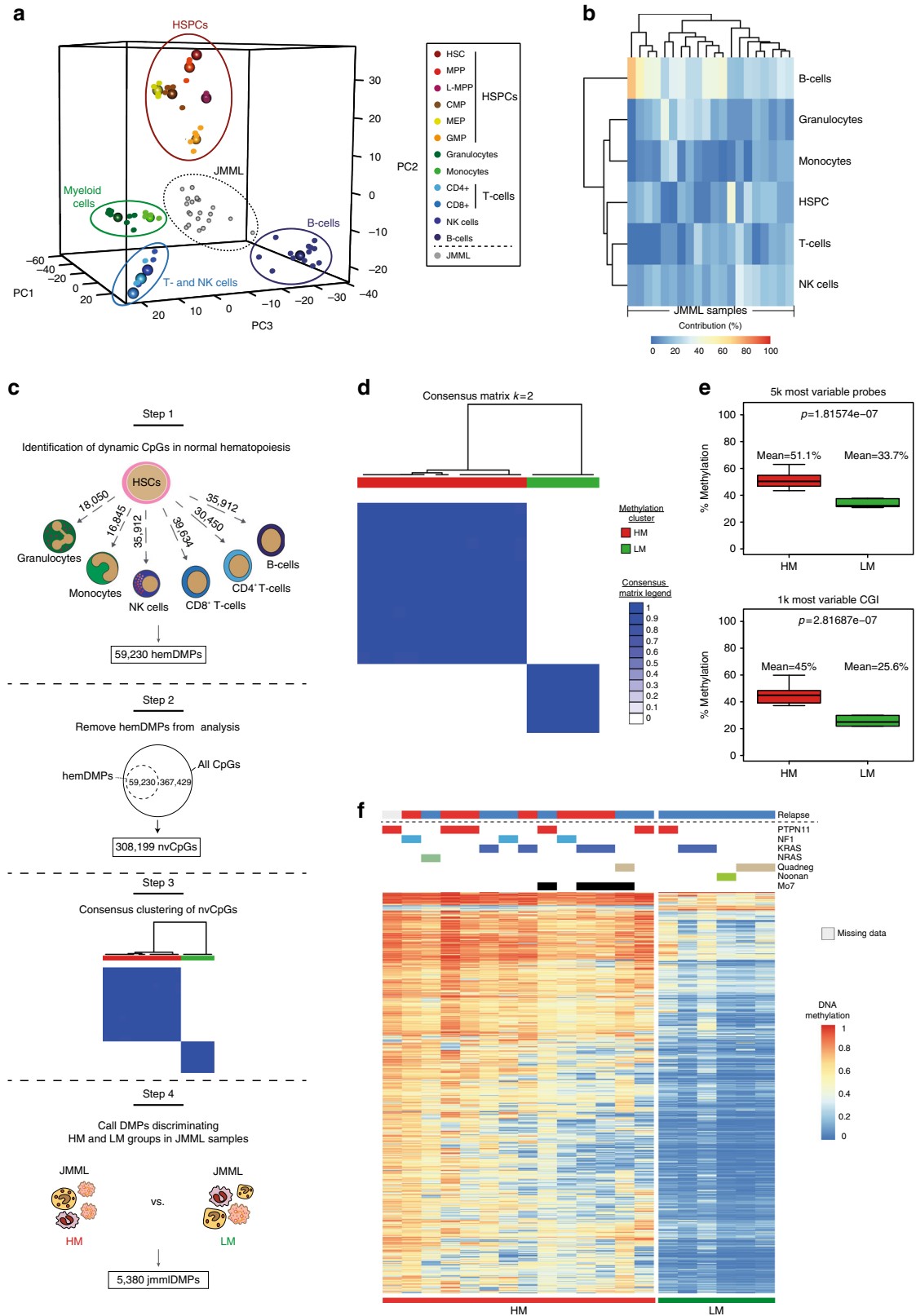

cluster was enriched for cases with chromosome 7 alterations, comprising 82% of all patients with this aberration. Moreover, a significant proportion of cases with *KRAS* mutations (65%), most of them co-occurring with chromosome 7 aberrations (92%), were assigned to the IM subgroup (Fig. 2 and Supplementary Table 2).

**DNA methylation is an independent prognostic factor in JMML.** Survival analysis of patients according to methylation group assignment revealed a strikingly inferior overall survival (OS) for the HM group as compared with the LM and IM groups ($n = 147$; log rank: $p < 0.01$; Fig. 3a). When considering only cases informative for outcome after HSCT and excluding patients with Noonan syndrome, *CBL* mutation or incomplete mutational analysis ($n = 92$; Supplementary Table 3), OS still tended to be inferior in the HM group, but the comparison failed to reach statistical significance (Supplementary Fig. 3a). Nevertheless, the strong association of methylation groups with known prognostic factors suggested that the methylation groups might reflect the risk of relapse after HSCT. As expected, the HM group showed a significantly higher cumulative incidence of relapse (CIR) than the other groups (Gray's test: $p < 0.01$; Fig. 3b). This difference was also reflected by poorer EFS in the HM group (log-rank test: $p = 0.09$ across all groups, $p = 0.03$ for HM vs. "other"), whereas treatment-related mortality (TRM) was not significantly different across methylation groups (Supplementary Fig. 3b, c). Of note, the selection of the stem cell donor, the type of preparative regimen, and the experience of the transplant center with tailoring of immunosuppressive therapy might be additional important factors influencing the risk of relapse after HSCT in JMML[3,27,28]. When restricting the analysis to patients who had an human leukocyte antigen (HLA)-identical sibling donor or a ≥ 9/10 HLA allele-level matched unrelated donor and who received a uniform preparative regimen and immunosuppressive therapy according to European Working Group of MDS in Childhood (EWOG) study recommendations ($n = 47$), the cumulative incidence of relapse was still significantly higher for patients in the HM group (Gray's test: $p = 0.03$; Supplementary Fig. 3d). This indicates that at least part of the increased risk is indeed attributable to the molecular disease group as defined by the DNA methylome.

Based on the finding that DNA methylation groups were significantly associated with risk of relapse, a cause-specific Cox model was fitted for which TRM was considered as competing risk. The model included the methylation group as well as known prognostic factors (i.e., age at diagnosis, sex, somatic *PTPN11*

mutation, and platelet count). Fetal hemoglobin levels were not included since data were missing in about 25% of patients. In this model, methylation group (HM vs. LM: RR 10.9 (1.8 – 66.2), HM vs. IM: RR 4.8 (1.4 – 17.2), IM vs. LM: RR 2.2 (0.4 – 11.2); $p = 0.01$, Wald's test) and *PTPN11* mutation status (PTPN11-mutant vs. other: RR 3.3 (1.2 – 8.9); $p = 0.02$, Wald's test) were identified as prognostic factors for CIR.

Together, the evaluation of clinical information further supports the hypothesis that the DNA methylation groups identify biologically distinct subgroups in JMML.

**RAS pathway mutations do not explain methylation groups.** The observation that methylation groups show enrichment for distinct gene mutations raises the possibility that distinct RAS pathway mutations drive the identified DNA methylation patterns. To test this hypothesis, DMPs were called separately for each genetically defined JMML group. Specific DMPs were identified for patients with Noonan syndrome, *CBL* syndrome, somatic *PTPN11*-, and somatic *KRAS* mutations (Noonan: 1,313; *CBL*: 279; *PTPN11*: 514; *KRAS*: 122), whereas no specific methylome pattern could be detected for patients with somatic *NRAS* mutations, *NF1* patients, and for quintuple-negative patients (6, 10, and 0 DMPs, respectively). Hierarchical clustering of all patients based on the "mutation-specific" DMPs did not provide a clear genotype-specific separation of patients for any of the DMP-classes, suggesting that the affected RAS pathway gene alone is not sufficient to explain the observed methylome patterns (Fig. 4a, b and Supplementary Fig. 4a, b).

To investigate whether differences in the mutated amino acid residues are associated with the methylation groups, a detailed analysis of affected amino acid residues and protein domains was performed. In *PTPN11*, the majority of mutations occur in the N-terminal SH2 domain (N-SH2, 82%). Interestingly, all of the 8 mutations occurring in the protein tyrosine phosphatase domain (PTP) were observed in either the IM (3/8) or the HM group (5/8; Fig. 4c). Glutamic acid at position 76 (E76), was the most frequently affected amino acid residue in the N-SH2 domain (35%). This residue showed a distinct methylation group distribution related to the type of amino acid exchange: The E76K mutation preferably occurred in the IM group (7/13), whereas the E76Q/G mutations were enriched in the HM group (6/7). The *KRAS* and *NRAS* mutations in JMML are predominantly G12D and G13D, respectively. In line with the enrichment for *KRAS* and *NRAS* mutations in the IM and LM groups, respectively, the *KRAS* G12D mutations were mainly found in the IM group, whereas *NRAS* G13D mutations were mainly found in the LM

**Fig. 1** Identification of JMML-specific aberrant DNA methylation patterns. **a** Three-dimensional principal component analysis (PCA) of DNA methylation dynamics across 12 normal hematopoietic cell types. JMML samples were projected as additional data points. CMP, common myeloid progenitors; GMP, granulocyte-macrophage progenitors; HSC, hematopoietic stem cells; HSPCs, hematopoietic stem and progenitor cells; L-MPP, lymphoid-primed multipotent progenitors; MEP, megakaryocyte-erythroid progenitors; MPP, multipotent progenitor cells; NK cells, natural killer cells. **b** Relative proportions of hematopoietic cell types in each sample from the discovery cohort ($n = 20$)[22]. **c** Strategy used to identify JMML-specific differentially methylated probes (jmmlDMPs). Step 1: differentially methylated probes (DMPs) exhibiting dynamic changes during normal hematopoietic differentiation were identified between HSCs and each of six differentiated blood cell types (granulocytes, monocytes, NK cells, CD8$^+$ T-cells, CD4$^+$ T-cells, and B-cells). Probes were considered as DMPs if the adjusted $p$-value was < 0.05 and the methylation difference (Δmeth) was ≥ 0.2. Step 2: 59,230 unique hematopoiesis-specific DMPs (hemDMPs) were identified and removed from further analysis, resulting in 308,199 CpGs that are non-variable in hematopoiesis (nvCpGs). Step 3: consensus clustering identified stable JMML subgroups, for which JMML-specific DMPs were identified using adjusted $p$-value < 0.05 and Δmeth ≥ 0.2 as filtering criteria. Step 4: identification of jmmlDMPs and clustering of JMML samples into subgroups. **d** Consensus clustering of the 5,000 most variable nvCpGs identified 2 stable groups ($k = 2$) separating JMML samples. The consensus matrix shows pairwise cluster assignment frequencies derived from 500 iterations based on Manhattan distance metric and Ward's linkage. Consensus values range from 0 (white) to 1 (dark blue). **e** Boxplots depicting the distribution of mean DNA methylation levels per JMML sample according to methylation group assignment across the 5,000 most variable CpG probes (top) and the 1,000 most variable CpG islands (CGI; bottom). Boxes represent the interquartile range and whiskers depict the minimum and maximum of the distribution. $P$-values are calculated using the two-sided unpaired Welch's $t$-test. **f** Hierarchical clustering of the 1,000 most variable jmmlDMPs using Manhattan distance metric and Ward's linkage. Samples (columns) are ordered according to consensus clustering results

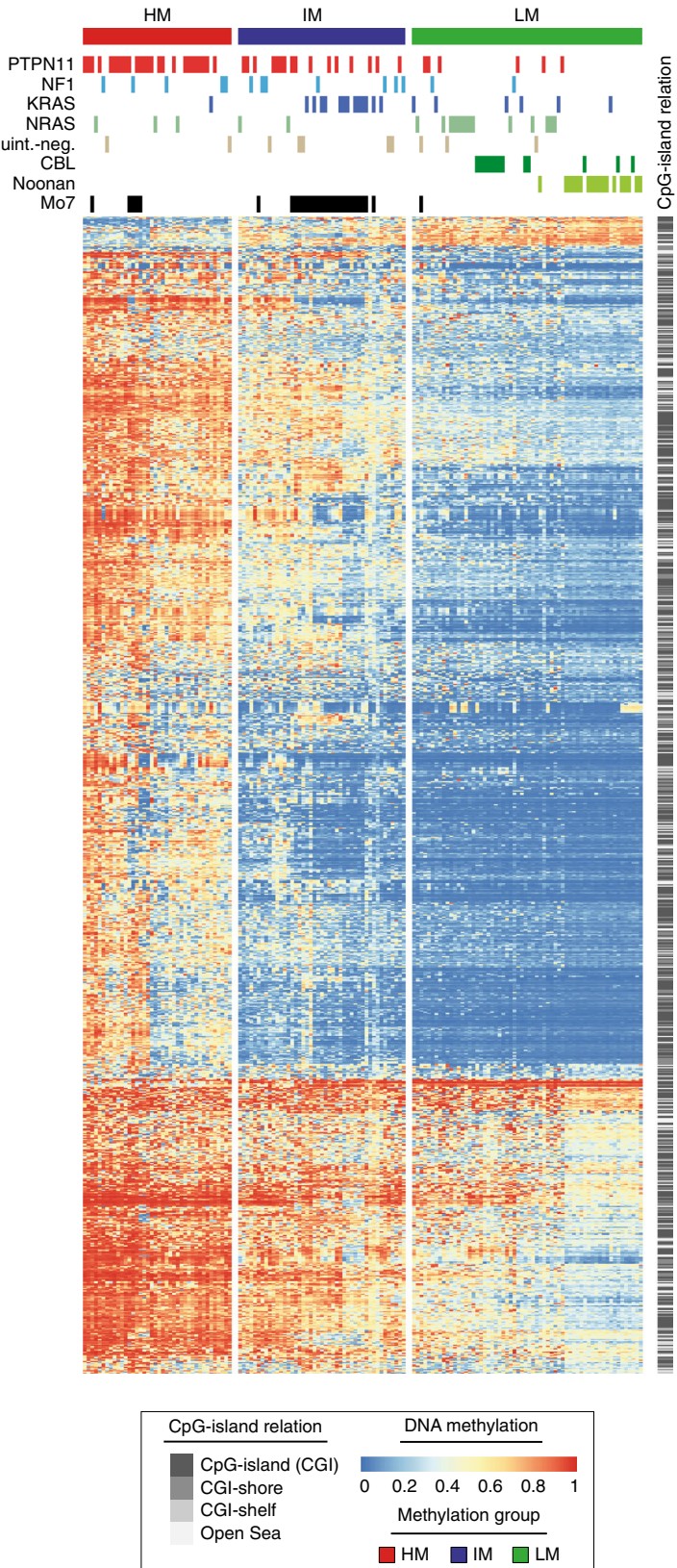

**Fig. 2** JMML-specific aberrant methylation patterns characterize three distinct JMML subgroups. Heatmap displaying beta values for the 1,000 most variable jmmlDMPs (rows) across all samples from the validation cohort ($n = 147$). Samples (columns) are ordered according to the consensus clustering results (Supplementary Fig. 2a). Clustering was performed using Manhattan distance and Ward's linkage. Clinical annotation for "genotype" (somatic mutations in *PTPN11*, *KRAS*, and *NRAS*, germline or somatic *CBL* mutations; clinical and/or molecular diagnosis of neurofibromatosis: NF1; quintuple-negative: quint.-neg.; Noonan: clinical and/or molecular diagnosis of Noonan syndrome) and karyotype is depicted on top of the heatmap. Relative DNA methylation levels are shown from light blue (0) to red (1), and localization of jmmlDMPs relative to CpG-islands is depicted on the right in gray scale

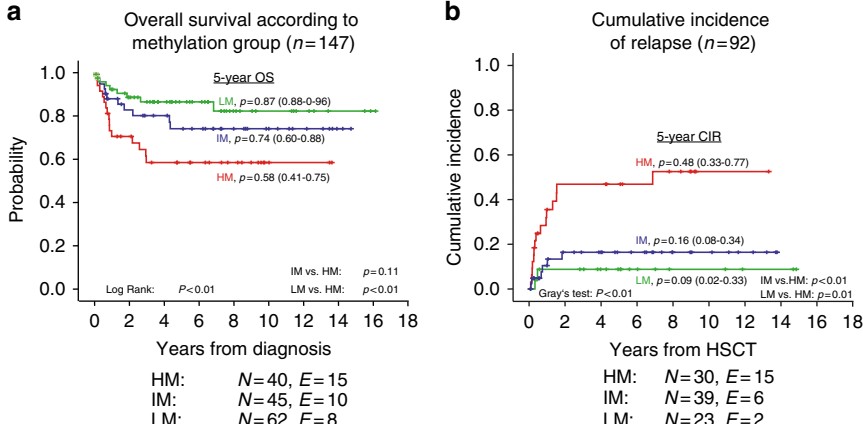

**Fig. 3** DNA methylation defines an aggressive JMML subgroup with high risk of relapse. Kaplan–Meier curves showing the clinical outcome of JMML patients stratified for methylation subgroups. HM: red curve, IM: blue curve, and LM: green curve. At the bottom of each graph the numbers of individuals at risk (N) and the numbers of events (E) are summarized according to methylation group. The curve labels represent the estimates for 5-year overall survival **a** and 5-year cumulative incidence of relapse **b** and the 95% confidence interval of the estimate. **a** Overall survival (OS) from diagnosis for the entire validation cohort (n = 147). Statistical significance was tested using log-rank test. **b** Cumulative incidence of relapse (CIR) for all patients with complete mutation analysis, who received HSCT (n = 92) and who did not have a diagnosis of Noonan syndrome or CBL syndrome (please refer to Supplementary Tables 2 and 3, and to Supplementary Data 3 for further information on patient characteristics). Statistical significance was tested using Gray's test

group. Although the numbers are too small to be of statistical significance, it is interesting to mention that 2/3 KRAS G12V and 2/3 NRAS G13R mutated cases were assigned to the HM group, which could potentially indicate distinct oncogenic potential of these mutations (Fig. 4d, e).

**JMML-specific DMPs are enriched for PRC2 and RAS target genes.** Investigation of the genomic distribution of jmmlDMPs revealed a strong enrichment for CpG islands, a depletion of non-coding RNAs and repetitive elements, and a depletion of intronic regions (Fig. 5a and Supplementary Fig. 5a, b). This supports the finding that the strongest methylation differences between HM and LM JMML were observed in CpG island probes (Fig. 1e) and is also compatible with a CIMP phenotype, suggesting deregulation of the epigenetic machinery.

Chromatin state and histone mark enrichment analysis using ENCODE data sets showed strong overrepresentation of jmmlDMPs in repressed chromatin (Fig. 5b and Supplementary Fig. 5c). In addition, jmmlDMPs were enriched for regions representing poised promoters, suggesting involvement of developmental pathways in JMML pathogenesis (Fig. 5b and Supplementary Fig. 5c). In line with these findings, gene set enrichment analysis (GSEA) showed a highly significant enrichment of regions decorated with H3K27me3, either alone or in combination with H3K4me2/3, and of regions bound by PRC2 components including EED and SUZ12 (Supplementary Fig. 5d). PRC2 is a chromatin-modifying complex mediating transcriptional repression and this complex has been shown to be frequently altered by mutations or deletions in JMML[10,11]. Most importantly, GSEA of those CpGs that exhibited the strongest deregulation in HM JMML as compared to both the LM and IM samples, revealed significant enrichment of genes associated with oncogenic RAS signaling (Fig. 5c). Together, these data imply that both aberrant function of the PRC2 complex as well as strong activation of RAS-signaling cooperate in the pathogenesis of JMML and the establishment of aberrant methylation patterns.

**The mutational signature correlates with methylome groups.** The mutational status of RAS signaling pathway genes assessed during clinical work-up did not provide a mechanistic

explanation for the JMML methylation groups observed. In search for secondary events that might explain our findings, integrative analysis of genetic and epigenetic events was performed in 50 patients of whom both methylome and exome-sequencing data sets were available. In addition to the classical JMML-associated mutations affecting RAS-pathway genes (PTPN11, NF1, KRAS, NRAS, and CBL), exome sequencing detected frequent mutations in JAK3 (10/50, 20%), SETBP1 (6/50, 12%), TET1 (5/50, 10%), ASXL1 (4/50, 8%), TET3 (4/50, 8%), RUNX1 (3/50, 6%), and TET2 (3/50, 6%). Copy-number alterations were frequently observed for EZH2 (12/50, 24%), NF1 (4/50, 8%), SUZ12 (4/50, 8%), and JAK3 (2/50, 4%; Supplementary Fig. 5e). Integration with methylome groups identified genes that were predominantly altered either in the HM and/or in the IM group (HM: ASXL1, RUNX1, SUZ12, TET1; HM & IM: EZH2, JAK3, SETBP1, and TET3), many of which are known epigenetic modifiers including members of the PRC2 complex or genes implicated in RAS-RAF-MEK-ERK pathway activation. The majority of JAK3 alterations occur in patients assigned to the HM or IM groups (9/11) and almost all co-occurred with PTPN11 mutations (7/11) or NF1 alterations (2/11). The remaining two patients with JAK3 mutations that were assigned to the LM group both had co-occurring NRAS mutations (Fig. 5d). Overall, both the HM and the IM groups were associated with a higher number of genetic alterations as compared to the LM group (HM vs. LM: 52 vs. 22, p < 0.001; IM vs. LM: 51 vs. 22, p < 0.001; Wilcoxon's test with continuity correction) and also showed higher frequencies of co-occurring mutations affecting the RAS-RAF-MEK-ERK pathway (Fig. 5e). Focusing on the most frequently altered genes, it became evident that a combination of these gene alterations almost perfectly discriminates the HM and IM samples from the LM samples.

**Hypermethylation is associated with activated RAS signaling.** Having detected an enrichment of gene sets associated with oncogenic RAS-signaling specifically in the HM group (Fig. 5c) and an increased co-mutation frequency of genes affecting the RAS signaling pathway (Fig. 5d, e), we investigated gene expression data sets available from a subset of samples from our discovery cohort. Promoter methylation levels globally showed

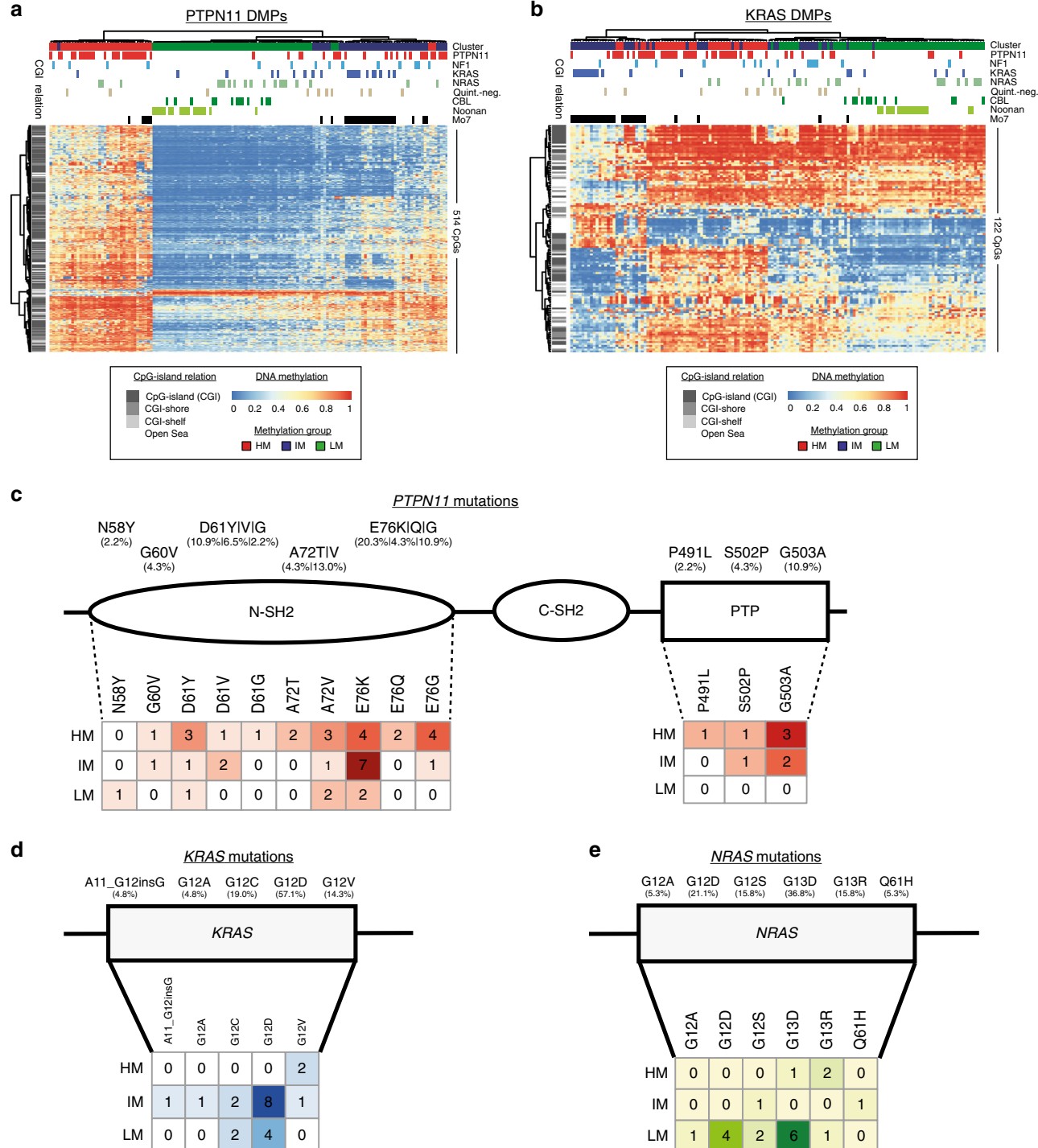

**Fig. 4** RAS pathway mutation patterns and their association with JMML methylation subgroups. Genotype-specific DMPs were called for cases with somatic *PTPN11* or *KRAS* mutations. **a**, **b** Unsupervised clustering of *PTPN11*-specific **a** and *KRAS*-specific **b** DMPs in all patients from the validation cohort (*n* = 147). Clinical annotation for "genotype" (somatic mutations in *PTPN11*, *KRAS*, and *NRAS*, germline or somatic *CBL* mutations; clinical diagnosis of neurofibromatosis: NF1; quintuple-negative: quint.-neg.; Noonan: clinical diagnosis of Noonan syndrome) and karyotype is depicted on top of the heatmap. DNA methylation levels are shown from light blue (0) to red (1). **c**, **d** Distribution of mutations in *PTPN11* **c**, *KRAS* **d** and *NRAS* **e** and their association with methylation subgroups. Numbers in the mutation matrices and color shading indicate the number of mutated cases per position and methylation group

inverse correlation with gene expression levels, with lowly expressed genes having the highest promoter methylation levels; nevertheless, obvious JMML subgroups could not be detected based on unsupervised hierarchical clustering of the most variably expressed genes (Supplementary Fig. 6a, b). Correlation of aberrantly methylated jmmlDMPs with the expression levels of corresponding genes across the JMML methylation subgroups

demonstrated a trend towards inverse correlation of gene expression with DNA methylation, which is in line with previous findings for correlation of gene expression and DNA methylation in normal blood cell types (Supplementary Fig. 6c)[29–31]. GSEA of the ranked gene list, revealed enrichment of genes known to be silenced by KRAS signaling as the only statistically significant hit (Supplementary Fig. 6d). This increase in RAS-mediated gene

silencing in the HM subgroup could potentially be attributed to differences in expression levels of genes regulating the RAS signaling pathway or, alternatively, to deregulation of epigenetic regulators. Interrogation of the gene expression data set revealed a significant upregulation of *PTPN11* expression in the HM group as compared with the LM group coinciding with downregulation

of the RAS signaling modulators *AKAP12* and *RASSF1* (Fig. 5f and Supplementary Fig. 5f). Of note, no significant deregulation was observed for other frequently mutated RAS pathway genes (i.e., *KRAS*, *NRAS*, *NF1*, and *CBL*, Supplementary Fig. 5f). Next, the expression of genes that have been implicated in RAS-mediated epigenetic silencing (*DNMT1*, *DNMT3A*,

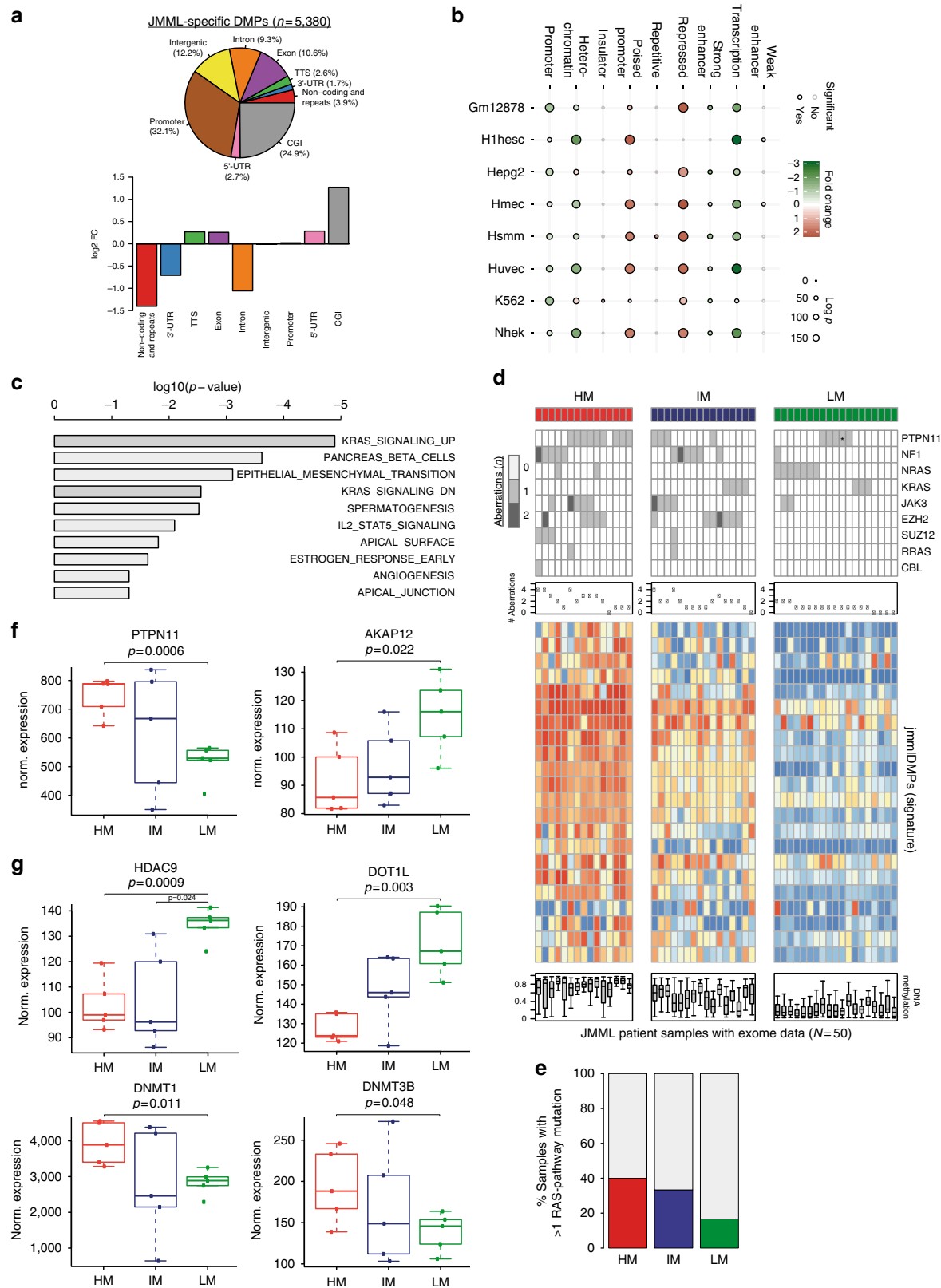

*DNMT3B*, *DOT1L*, *EED*, *EZH2*, *HDAC9*, *SUZ12*, and *TET1-3*) was examined (Fig. 5g and Supplementary Fig. 5f)[18,32–35]. Both *DOT1L* and *HDAC9* transcripts were downregulated, whereas *DNMT1* and *DNMT3B* were upregulated in the HM group. Interestingly, differences in promoter methylation that would go along with the observed expression changes were only found for *AKAP12* and *HDAC9* (Supplementary Fig. 7). An increase in promoter methylation was also observed for *TET1* but the corresponding gene expression did not show significant changes in the present data set (Supplementary Figs. 5f and 7). None of the remaining candidate genes presented detectable changes in promoter methylation levels at the CpG sites tested with the HumanMethylation450 Bead Chip Array (Supplementary Fig. 7). The lack of DNA methylation changes in many of our transcriptionally deregulated candidate genes could be due to insufficient coverage of cis-regulatory CpG sites on the HumanMethylation450 Bead Chip Array, but it is also possible that transcriptional deregulation is mediated by other (epigenetic) mechanisms.

Together, activation of the RAS pathway by gene mutations might be further modulated by overexpression of *PTPN11* and simultaneous downregulation of negative RAS signaling modulators in HM JMML patients. In addition, the HM group exhibits small but significant expression changes in components of the epigenetic machinery including upregulation of *DNMT1* and *DNMT3B*. However, if the expression changes observed here are sufficient to drive CIMP or if additional secondary regulatory events affect the activity of epigenetic factors needs to be examined in future studies, in order to provide mechanistic insights into how the CIMP in JMML is established.

## Discussion

Molecular classification of cancer based on DNA methylomes has revolutionized the identification of tumor subclasses both in neurooncology[36–39] and in chronic lymphocytic leukemia[40,41]. Here we used an integrative approach and identified three clinically relevant JMML subgroups with distinct molecular genetic patterns (Fig. 6). The HM subgroup is enriched for patients with somatic *PTPN11* mutations and includes cases with poor clinical outcome. The LM subgroup encompasses all patients with *CBL* syndrome and Noonan syndrome with MPD, as well as patients with somatic *NRAS* mutations and low-risk features. It is intriguing that the disorders in these patient groups, which are all characterized by a less aggressive clinical course, also share a highly similar methylome. The IM group is enriched for cases with monosomy 7 and somatic *KRAS* mutations, suggesting that the joint occurrence of these genetic alterations drives a specific epigenetic profile in JMML. Although the DNA methylation subgroups correlate with known prognostic factors in JMML (i.e., age at diagnosis, platelet count, genotype, and elevated HbF levels)[2,15,42], only the methylation subgroup and, to a lesser extent, the presence of a somatic *PTPN11* mutation were significant predictors for risk of relapse in a multivariate Cox-regression model. This finding is of high conceptual relevance for JMML, as currently available risk stratification strategies rely on clinical parameters for which their individual weight remains uncertain. Similar to the approach proposed for pediatric brain tumors, we anticipate that classification based on underlying molecular defects, including DNA methylation patterns, will improve prospective patient stratification also in JMML.

JMML is characterized by the presence of mutations activating the RAS signaling pathway in about 90% of cases and these mutations are mostly mutually exclusive[15,43]. Our present data extends these findings and demonstrates that DNA hypermethylation is a hallmark of aggressive JMML. Despite being a genome-wide phenomenon, hypermethylation in JMML is most pronounced in the context of CpG-rich regions such as, e.g., CpG islands. This finding is similar to DNA hypermethylation phenotypes found in other cancers that were described as CIMP[23–25]. Early reports had implicated RAS signaling in the regulation of DNA methylation[16]. Meanwhile, there is accumulating evidence that oncogenic KRAS-signaling mediates DNA hypermethylation and CIMP[17,18,44]. So far, more than 30 genes have been shown to be required to establish RAS-mediated CIMP, many of which seem to act in a tissue-specific manner. The most recurrently identified genes in this context are *DNMTs*, *EED*, *EZH2*, and *BMI1*[16–18,45].

In recent work with murine cells, oncogenic Ras signaling has been shown to induce transcriptional changes leading to an accumulation of H3K27me3 and to mediate upregulation of Dnmt1 and Dnmt3a expression, which provides a molecular link for RAS-mediated transcriptional silencing and DNA hypermethylation[45–47]. Furthermore, it has been shown that phosphorylated c-Jun, which is regulated in a RAS-dependent manner, upregulates DNMT1 expression and causes DNA hypermethylation[48]. Oncogenic KRAS is also able to suppress the expression of TET1 in an ERK-dependent manner, thereby contributing to DNA hypermethylation[32]. Together, there is now substantial evidence that oncogenic activation of the RAS-RAF-MEK-ERK pathway mediates epigenetic remodeling and contributes to disease pathogenesis in different tissue contexts. Our present data on JMML suggest that RAS-activating mutations in different genes might have distinct effects on epigenome remodeling correlating with disease aggressiveness. Along these lines, recent work evaluating the role of gene mutations on the prognosis of myelodysplastic syndromes demonstrated that patients with mutations in *KRAS* or *NF1* show worse OS than patients

**Fig. 5** Aberrant DNA methylation patterns are associated with signaling pathway activation and overexpression of DNMTs. **a** Distribution of jmmlDMPs across distinct genomic features (top). The bottom panel shows the enrichment analysis of genomic features in jmmlDMPs as compared with background probes. **b** Bubble chart depicting the enrichment (red) or depletion (green) of chromatin states in jmmlDMPs across eight different cell lines. The dot colors represent the logarithmic fold change and the dot size indicates the log(*p*)-value for each enrichment. The outline colors indicate statistical significance (black: significant, gray: not significant). **c** Results of gene set enrichment analysis using the molecular signature database (MSigDB; http://software.broadinstitute.org/gsea/msigdb/index.jsp)[60]. The bar plot depicts the top ten gene sets enriched in the HM JMML subgroup based on *p*-values from the hypergeometric distribution. **d** Integrative analysis of mutations, copy-number alterations and methylome patterns in all JMML patients for whom both exome-seq and methylome data were available (*n* = 50). Depicted are events in genes known to be involved in RAS and/or STAT signaling pathway activation and events affecting PRC2-related genes. Methylation patterns are depicted for 19 signature CpG probes that were selected for their ability to separate JMML subgroups using a cluster prediction model. *Presence of a germline *PTPN11* (p.73 T > I) mutation in the context of Noonan syndrome. **e** Bar chart depicting the frequency of tumor samples with > 1 mutations activating the RAS/STAT pathways according to methylation subgroup. **f**, **g** Expression of RAS signaling genes **f** and of genes involved in epigenetic regulation **g**. Depicted are quantile normalized gene expression microarray data from 15 JMML patient samples from the discovery cohort for whom RNA of sufficient quality was available. For this analysis, methylation groups were re-assigned based on the three group methylation classifier. The boxes represent the interquartile range and whiskers depict the minimum and maximum of the distribution not considering outliers. Two-sided unpaired Welch's *t*-test was used to test for expression differences between HM or IM vs. LM subgroups

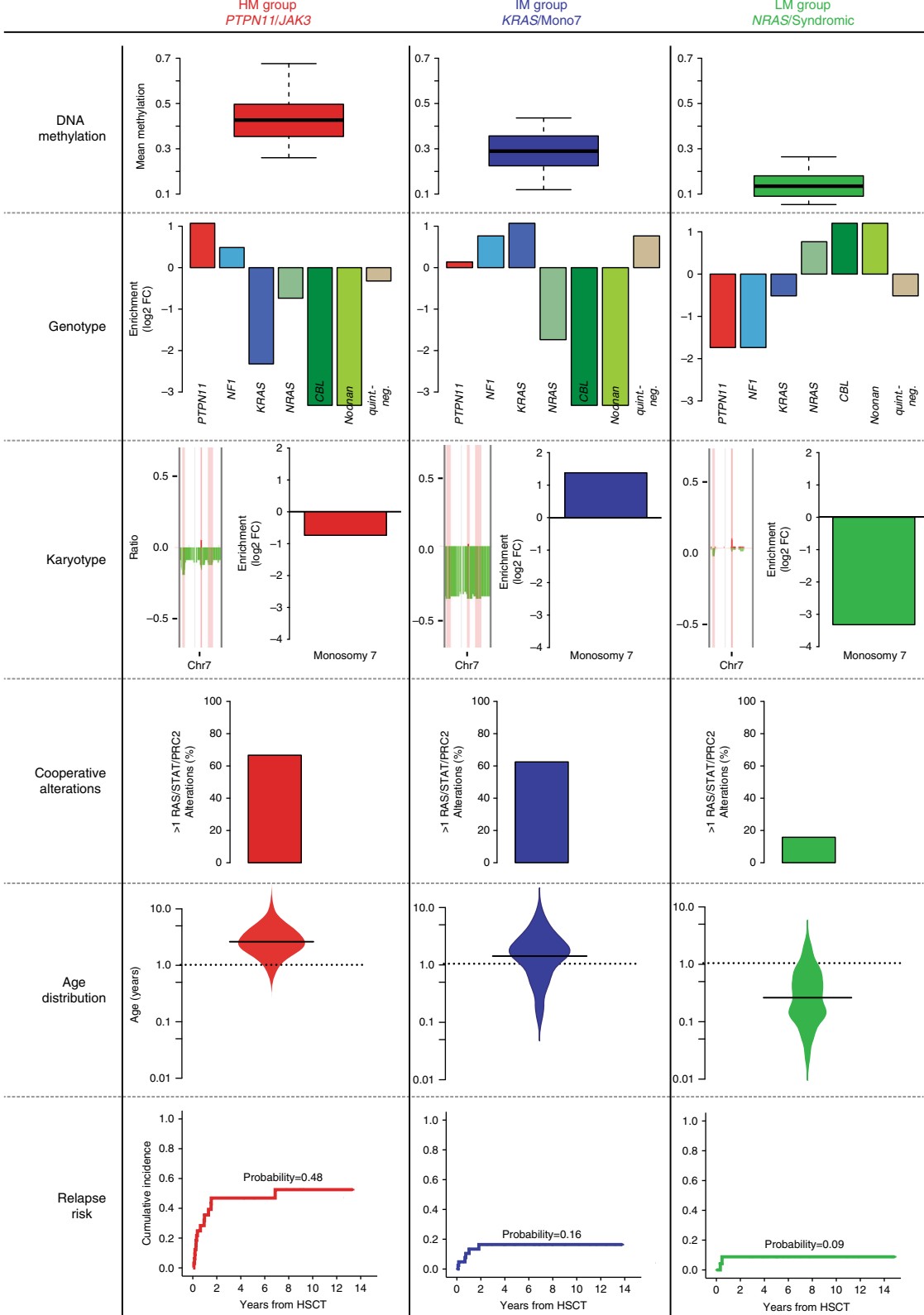

**Fig. 6** Clinical and molecular features associated with JMML methylome subgroups. This figure summarizes the clinical and molecular features of the JMML methylome subgroups. Average DNA methylation levels in jmmlDMPs (DNA methylation), enrichment of RAS-pathway mutations (Genotype), enrichment of monosomy 7 (Karyotype), frequency of > 1 RAS/STAT/PRC2 alterations (Cooperative Alterations), age at diagnosis (Age Distribution), and relapse risk (Relapse Risk)

with *NRAS* or *CBL* mutations[49]. It is likely to be that these distinct effects are mediated by quantitative and qualitative differences in the activation of signaling pathways. Interestingly, in this context, a recent study has demonstrated that *PTPN11*-E76K mutations located in the N-SH2 domain confer RAS-independent activation of the PI3K/mTOR pathway, suggesting aberrant recruitment of binding partners[50]. Furthermore, we have shown that secondary mutations providing additional activation of RAS-signaling and other signaling pathways are frequent in high-risk JMML. One might speculate that these secondary mutations contribute to transcriptional deregulation of *DNMT1* and thus further augment the extent of epigenetic remodeling. Alternatively, pre-existing epigenetic alterations might provide a fertile ground for malignant transformation following single or few genetic hits. This sequence of events has been shown recently in the context of cigarette smoke-induced lung cancer where DNA hypermethylation of PRC2 target genes sensitizes bronchial epithelial cells to single-step transformation by mutant *KRAS*[51].

In summary, our data provide strong evidence for the existence of three methylation subgroups in JMML, which are characterized by distinct clinical and biological features and provide the rationale for future work to dissect the molecular mechanisms underlying the methylation subgroups in JMML.

## Methods

**Patient samples.** The discovery cohort consisted of a series of mononuclear cell samples from JMML patients ($n = 19$) and from one patient with Noonan syndrome and MPD purified from splenectomy preparations and cryopreserved at the biorepository of the Division of Pediatric Hematology and Oncology, University Medical Center Freiburg, Germany. The selection of samples for the discovery series was solely based on availability of sufficient cell numbers. Basic clinical information for the discovery cohort patients is summarized in Supplementary Table 1. The validation cohort consisted of 147 consecutive patients diagnosed with JMML or Noonan syndrome associated MPD, who were registered to the EWOG-MDS studies EWOG-MDS98 and EWOG-MDS2006 (NCT00047268 and NCT00662090; www.clinicaltrials.gov). The patient samples for this cohort were chosen based on availability ($\geq 1$ μg DNA) and quality (no degradation) of leukemic granulocyte DNA purified from bone marrow or peripheral blood and time-point of sample collection from peripheral blood or bone marrow ($\leq 6$ months after diagnosis but before HSCT). Information on JMML genotype/mutation category was provided by the Coordinating Study Center of EWOG-MDS, as previously determined by a synopsis of clinical features (neurofibromatosis type 1, Noonan syndrome, CBL syndrome) and molecular work-up (Sanger sequencing of *PTPN11* exons 3, 4, 8, and 13, *KRAS* exons 2 and 3, *NRAS* exons 2 and 3, and *CBL* exons 8 and 9 in leukemic (somatic) and germline material). Parents or legal guardians of all patients had provided informed consent to scientific use of patient materials in accordance with the Declaration of Helsinki. The collection and storage of patient materials was approved by the institutional review board of each participating center. Patient characteristics are summarized in Supplementary Tables 2 and 3, and in Supplementary Data 3.

**Isolation of normal B-cells and granulocytes.** B-cells (CD19+) and granulocytes (CD15+) were isolated from peripheral blood of healthy donors using magnetic cell separation according to the manufacturer's instructions (MACS, Miltenyi Biotec).

**Published data sets used in this study.** We used publicly available methylome data sets from normal HPSCs[52] and differentiated blood cells[53] as a reference.

**Genome-wide DNA methylation analysis.** Genomic DNA (100 – 200 ng) was analyzed by the DKFZ Genomics and Proteomics Core Facility for DNA methylation profiling using the Infinium HumanMethylation450 Bead Chip Array (Illumina, San Diego, CA, USA). Beta-mixture quantile normalization with background correction based on a normal-exponential model using out-of-band intensities ("methylumi.noob") was used to normalize idat files using the RnBeads package[54]. The RnBeads options 'filtering.snp = "5"' and 'filtering.sex.chromosomes. removal = TRUE' were used to filter out all CpG-probes within 5 bp of a known single-nucleotide polymorphism (SNP) and all probes recognizing regions on the sex chromosomes. Additional SNP filtering was performed using the "filtering.blacklist" option to filter out all CpG-probes within 3 bp of any SNP in dbSNP version 146. After these filtering steps, a total of 367.429 probes were retained. All samples, including HSPCs and differentiated blood cells from healthy individuals, as well as all JMML samples, were processed simultaneously using the parameters mentioned above.

**Analysis of copy number variations using 450 K data.** Copy number variations were called for all JMML samples from the 450 K raw-intensity data using the bioconductor package "conumee"[55]. The Genomic Identification of Significant Targets in Cancer (GISTIC2) method was subsequently used to identify significant and recurrent copy number alterations based on the conumee raw copy number output[56].

**Inference of cell-type composition.** Relative proportions of different cell types in the JMML samples were estimated using the RnBeads option for cell type inference that implements the algorithm originally developed by Houseman et al[22].

**Principal component analysis of hematopoiesis.** Principal component analysis of methylation dynamics in normal hematopoiesis was performed using the R package FactoMineR[57]. The top 5,000 most variable CpGs across the six HSPC populations (HSC, MPP, L-MPP, MEP, CMP, and GMP) were used to calculate the principal components 1–5 for all normal hematopoietic cell types. JMML samples from the discovery cohort were used as supplementary variables.

**Cluster analysis of JMML samples.** Consensus clustering was performed with 500 bootstrap iterations as implemented in the ConsensusClusterPlus package[58]. Hierarchical clustering was performed based on Manhattan distance and Ward's linkage (ward.D2) using the 5,000 and 1,000 most variable probes as indicated.

**Differential DNA methylation calling.** Differential methylation analysis between sample groups (i.e., HSCs vs. committed blood cell populations and between JMML sample groups) was performed using the rnb.run.differential function from the RnBeads package[54]. Probes with false discovery rate-corrected *p*-value < 0.05 and a methylation difference (Δmeth) ≥ 0.2 were considered as DMPs. All analyses were carried out in an R computing environment.

**Annotation and enrichment analysis.** DMPs were annotated to genomic regions based on the NCBI reference genome (hg19) and enrichment of genomic regions was calculated using HOMER version 3.9[59]. Functional annotation of jmmlDMPs was performed using GSEA on gene sets available from the Molecular Signature Database[60,61].

The results of 15-state ChromHMM models trained on eight cell lines were downloaded from ENCODE. States were then combined to nine meta-states[62]. ChIP-seq tracks for 13 cell lines in BED format were downloaded from ENCODE[62]. All jmmlDMPs were used as foreground and the remaining non-differentially methylated nvCpG sites form a background set. Enrichment analysis was performed independently for every pair (state/cell line) on the foreground set as compared to the background set. Logarithmic fold change values were calculated as log2(OF/OB), where OF is the fraction of probes in the foreground set overlapping with the chromatin state type of interest and OB is the same metric for the background set. *P*-values were obtained using Fisher's exact test and adjusted for multiple testing using the Benjamini–Hochberg method. The significance threshold applied was 0.01.

**Exome sequencing of JMML samples.** Sufficient amounts of DNA were available for exome-sequencing from 50 JMML patients (discovery cohort and validation cohort). Sequencing libraries were prepared at the DKFZ Genomics and Proteomics Core Facility using the Agilent "SureSelectXT Human all Exon V4" kit and subsequently sequenced on a HiSeq2000 instrument using the 100 bp paired-end mode. Alignment to the hs37d5 reference genome was performed using BWA-MEM[63]. SAMtools/BCFtools (version 0.1.19) were used for single nucleotide variants (SNV) calling[64]. SNV found with high frequency in other mutation databases (i.e., 'common = 1' tag in dbSNP or > 1% frequency in ExAC 0.3.1) were filtered out and only high-confidence mutations in the coding regions were kept. Calling of small insertions/deletions (INDEL) was performed using Platypus 0.8.1 using the same filtering process as with the SNV calling[65]. Both SNV and INDEL calling were performed without paired germline control samples. The data were plotted as an OncoPrint using the ComplexHeatmap package[66]. Statistics between groups were performed using the non-parametric Wilcoxon's test.

**Cluster prediction model.** To prospectively classify JMML samples according to their methylation profiles, the 1,000 most variables jmmlDMPs mapping to CpG islands were used to develop a prediction model. Samples from the validation cohort were split into training and test groups in a 4:1 ratio. A multinomial lasso regression model was fit on the training data set using glmnet package version 2.0–5[67]. The penalty parameter was selected using 10-fold cross-validation. The model evaluation was done using a modified Cohen's kappa statistics on the test set as follows[68]:

$$K = \frac{\text{Acc} - \text{AccRand}}{1 - \text{AccRand}}$$

Accuracy (Acc) was calculated as a ratio of true positive and all samples. Random accuracy (AccRand) was calculated by taking the mean accuracy from 1,000 times random permutation of the predicted classes.

**Statistical analysis**. For this study, the database on patient outcome was locked 1 October 2016. OS was defined as the time from diagnosis to death or last follow-up. EFS was defined as the time from HSCT to treatment failure (i.e., death or leukemia relapse, whichever occurred first) or last follow-up. The Kaplan–Meier method was used to estimate survival rates and the two-sided log-rank test was employed to evaluate the equality of the survivorship functions in different subgroups. Time-to-event outcomes for relapse (CIR) and TRM were estimated using cumulative incidence curves with relapse and TRM as reciprocal competing risks. Differences in the cumulative incidence functions among groups were compared using Gray's test. $\chi^2$-test was used to examine the statistical significance of an association between categorized factors. In the case of a $2 \times 2$ contingency table, Fisher's exact test was calculated. Median values and ranges were reported and nonparametric statistics were used to test for differences in continuous variables among methylation groups (Mann–Whitney $U$-test). For multivariate analyses, the Cox proportional hazard regression model was used, including the methylation group as well as known prognostic factors (i.e., age at diagnosis, sex, somatic *PTPN11* mutation, and platelet count). All $p$-values were two-sided and values < 0.05 were considered statistically significant. $P$-values > 0.1 were reported as nonsignificant, whereas those between 0.05 and 0.1 were reported in detail. SPSS for Windows 22.0.0 (IBM Corp.) and NCSS 2004 (Number Cruncher Statistical Systems) were used for the statistical analysis of the data.

**Gene expression microarray analysis**. Total RNA was submitted to the DKFZ Genomics and Proteomics Core Facility where Illumina HumanHT-12 v4 Expression BeadChips were hybridized and scanned according to the manufacturer's recommendations. After the initial quality checks, the missing data was imputed using nearest neighbor averaging as implemented in the impute package[69]. During the preprocessing, robust spline normalization and vst transformation were applied, both as implemented in the lumi package[70]. All analysis was performed using R software.

**Data availability**. All methylome, exome-seq and gene expression data have been deposited in ERA https://www.ebi.ac.uk/ega/home (study number: EGAS00001002511).

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

## Acknowledgements

We thank all members of the Division of Epigenomics and Cancer Risk Factors (DKFZ) for thoughtful discussions related to this study and we thank Monika Helf and Oliver Mücke for excellent technical support. We also thank the Microarray unit and the High Throughput Sequencing unit of the Genomics and Proteomics Core Facility, German Cancer Research Center (DKFZ), for providing the Illumina HumanMethylation450 arrays, the Illumina HumanHT-12 v4 Expression BeadChip arrays and related services, as well as for excellent support with exome sequencing services. This work was supported by project funding from the German José Carreras Leukemia Foundation (DJCLS) to C. P., D.B.L., and C.F. (project: DJCLS R 15/01), and from the German Research Foundation (DFG) to C.F. (projects: CRC992-C05 and FL345/4-2), and to H.B. (EXC 306) and M.B. (CRC 850). M.B. is funded by the German Federal Ministry of Education and Research within the framework of the e:Med research and funding concept, DeCaRe (FKZ 01ZX1409B). U.K. was supported by a joint German Academic Exchange Service (DAAD) and Turkish Education Foundation (TEV) master scholarship. This work was further supported by the German Cancer Consortium (DKTK).

## Author contributions

D.B.L., T.W., C.F., and C. Plass designed and coordinated the study. D.B.L., T.W., R.T., J. Y., D.B., Z.G., J.P., H.B., M.B., Y.A., and M. Schlesner performed data analysis and interpreted the data. M.W. and P.N. performed statistical analysis and interpreted the data. A.F., B.S., M.W., A.Y., R.C., M.L., M.H., M. Schönung, U.K., J.L., J.W., C. Pabst, S. G., A.C., B.D.M., M.D., H.H., F.L., R.M., M. Schmugge, O.S., J.S., M.U., M.M.v.d.H.-E., C. F., and C.N. contributed patient samples, and/or reagents and materials. D.B.L., T.W., and C. Plass wrote the first draft of the manuscript. D.B.L., C.F., and C. Plass jointly directed and supervised the research. All coauthors contributed to the final version of the manuscript.

## Additional information

**Competing interests:** The authors declare no competing financial interests.

Daniel B. Lipka[1,2,3], Tania Witte[1,4], Reka Toth[5], Jing Yang[6], Manuel Wiesenfarth[7], Peter Nöllke[8], Alexandra Fischer[8], David Brocks[4], Zuguang Gu[6], Jeongbin Park[6], Brigitte Strahm[8], Marcin Wlodarski[8,12], Ayami Yoshimi[8], Rainer Claus[9], Michael Lübbert[9], Hauke Busch[10,11], Melanie Boerries[10,12,13], Mark Hartmann[1], Maximilian Schönung[1], Umut Kilik[1], Jens Langstein[1], Justyna A. Wierzbinska[1,4], Caroline Pabst[14],

Swati Garg[14], Albert Catalá [15], Barbara De Moerloose[16], Michael Dworzak[17], Henrik Hasle [18], Franco Locatelli[19], Riccardo Masetti [20], Markus Schmugge[21], Owen Smith[22], Jan Stary[23], Marek Ussowicz[24], Marry M. van den Heuvel-Eibrink[25], Yassen Assenov[5], Matthias Schlesner [6,26], Charlotte Niemeyer[8,12], Christian Flotho[8,12] & Christoph Plass [4,27]

[1]Regulation of Cellular Differentiation Group, Division of Epigenomics and Cancer Risk Factors, German Cancer Research Center (DKFZ), INF 280, 69120 Heidelberg, Germany. [2]Department of Hematology and Oncology, Medical Center, Otto-von-Guericke-University, Leipziger Strasse 44, 39120 Magdeburg, Germany. [3]Health Campus Immunology, Infectiology and Inflammation, Otto-von-Guericke-University, Leipziger Strasse 44, 39120 Magdeburg, Germany. [4]Cancer Epigenetics Group, Division of Epigenomics and Cancer Risk Factors, German Cancer Research Center (DKFZ), INF 280, 69120 Heidelberg, Germany. [5]Computational Epigenomics Group, Division of Epigenomics and Cancer Risk Factors, German Cancer Research Center (DKFZ), INF 280, 69120 Heidelberg, Germany. [6]Division of Theoretical Bioinformatics (B080), German Cancer Research Center (DKFZ), INF 280, 69120 Heidelberg, Germany. [7]Division of Biostatistics, German Cancer Research Center (DKFZ), INF 280, 69120 Heidelberg, Germany. [8]Division of Pediatric Hematology and Oncology, Department of Pediatrics and Adolescent Medicine Medical Center, Faculty of Medicine, University of Freiburg, Heiliggeiststrasse 1, 79106 Freiburg, Germany. [9]Division of Hematology, Oncology and Stem Cell Transplantation, University Medical Center, Hugstetter Strasse 55, 79106 Freiburg, Germany. [10]Institute of Molecular Medicine and Cell Research, University of Freiburg, Stefan-Meier-Strasse 17, 79104 Freiburg, Germany. [11]Lübeck Institute of Experimental Dermatology, University of Lübeck, Ratzeburger Allee 160, 23562 Lübeck, Germany. [12]German Cancer Consortium (DKTK), 79106 Freiburg, Germany. [13]German Cancer Research Center (DKFZ), 69120 Heidelberg, Germany. [14]Department of Hematology, Oncology and Rheumatology, Heidelberg University Hospital, INF 410, 69120 Heidelberg, Germany. [15]Department of Hematology and Oncology, Hospital Sant Joan de Déu, Passeig de Sant Joan de Déu, 2, 08950 Esplugues de Llobrega, Barcelona, Spain. [16]Department of Pediatric Hematology-Oncology and Stem Cell Transplantation, Ghent University Hospital, De Pintelaan 185, 9000 Ghent, Belgium. [17]St. Anna Children's Hospital and Children's Cancer Research Institute, Medical University of Vienna, Zimmermannplatz 10, 1090 Vienna, Austria. [18]Department of Pediatrics, Aarhus University Hospital Skejby, Palle Juul-Jensens Boulevard 82, 8200 Aarhus, Denmark. [19]Department of Pediatric Hematology and Oncology, Bambino Gesú Children's Hospital, University of Pavia, Piazza S. Onofrio 4, Rome 00165, Italy. [20]Department of Pediatric Oncology and Hematology, University of Bologna, Via Massarenti 11, 40138 Bologna, Italy. [21]Department of Hematology and Oncology, University Children's Hospital, Steinwiesstrasse 75, 8032 Zurich, Switzerland. [22]Department of Paediatric Oncology and Haematology, Our Lady's Children's Hospital Crumlin, Dublin 12, Ireland. [23]Department of Pediatric Hematology and Oncology, Charles University and University Hospital Motol, V Úvalu 84, 150 06 Prague 5, Czech Republic. [24]Department of Pediatric Hematology, Oncology and BMT, Wroclaw Medical University, ul. Borowska 213, 50-556 Wroclaw, Poland. [25]Princess Maxima Center for Pediatric Oncology, Lundlaan 6, 3584 EAUtrecht, The Netherlands. [26]Bioinformatics and Omics Data Analytics (B240), German Cancer Research Center (DKFZ), 69120 Heidelberg, Germany. [27]German Cancer Consortium (DKTK), 69120 Heidelberg, Germany. Daniel B. Lipka and Tania Witte contributed equally to this work. Christian Flotho and Christoph Plass jointly supervised this work.

