## [Peer Review File · Nature Communications]

Reviewers' comments:

Reviewer #1 Expert in leukaemia genetics:

The authors study the methylation patterns of DNA of patients with juvenile myelomonocytic leukemia (JMML). They hypothesized that DNA methylation profiling, alone or in combination with genetic alterations, could predict disease severity and outcome. DNA methylome analysis and mutation profiling was performed for 167 JMML samples. Clustering based on the variable CpG sites identified three clusters of patients (high, medium and low methylation), with the lowest methylation cluster having the highest rates of survival. The author thus demonstrated convincingly that methylation analysis has predictive value in JMML.

The authors look for the reason of the methylation changes, and propose a hypothesis, but fail to identify the cause of the methylation changes. The authors claim that differentially methylated sites were enriched for regions occupied by H3K27me3 or PRC2 and for genes associated with RAS signalling, but that part is relatively weak and only shown by indirect evidence.

Overall, the methylation data is nice and seems to predict clinical outcome very well. This is the strong part of the paper. The reason why the DNA methylation is high or low in the different patients, is less clear, and based on data that is not very strong or not supported by data in JMML. It would be good to tune this down or include more data.

Major remarks/questions:

1. Figure 1 demonstrates that a correction of the methylation data is required, for the various contributions of different cell types in each sample. Is this really needed? And how does the data look for JMML if no such correction is done. In the accompanying article (with some of the authors shared with this article), this seems not to be required.
2. Figure 5: methylation data is integrated with publicly available ChIP data for various histone marks (Encode data). This is of interest to generate a hypothesis, but would need to be confirmed by ChIP experiments with JMML samples. In the current version of the manuscript, the conclusions are highly speculative and not confirmed by ChIP-seq data for JMML. Similarly, figure 5H shows data on AML cell lines, which has little value. I would suggest to remove this part (figure 5) or to confirm the data in JMML samples with new experiments.
3. The DNA methylation profiles are clear, but have not been closely linked with expression data. Since DNA methylation is expected to influence expression, it would be a nice and essential

addition to include a pairwise comparison between DNA methylation changes and RNA expression levels based on RNA-seq data. In the current manuscript this is only shown for a few selected genes.

4. Related to the previous remark: if RNA expression data is used for clustering, would this also result in a similar division over 3 clusters, with similar predictive value for survival/relapse?

Minor remarks:

- Please explain abbreviations used, for example 'DMPs' is not explained as well as some other abbreviations.
- Figure 1B seems very black and white. There are no intermediate values. Is this correct?

Reviewer #2 Expert in leukaemia epigenetics:

In the present manuscript, the authors hypothesized that DNA methylation profiling, alone or in combination with genetic mutation profiling, might provide a molecular basis for JMML classification. With this approach the authors identified 3 JMML subgroups that showed unique molecular and clinical characteristics.

The study is clearly written and may lead to relevant prognostic data that might influence treatment of the JMML.

Here some suggestions to implement the preclinical and clinical value of the manuscript.

The authors indicate that up-regulation of DNMT1 and DNMT3B in the HM group is likely to be the molecular surrogate of the CIMP phenotype observed. The authors might include epigenome-based details on the cause of the identified up-regulation as well as they might verify using cell models if using chromatin modulators it is possible to re-equilibrate DNMTs expression of activity.

The authors also indicate that leukemias carrying RAS-pathway mutations exhibit methylation patterns akin to those observed in JMML (Fig. 5H). Could the authors use those cells to include some combo-treatments to verify whether targeting RAS (or the other) identified pathways together with potential chromatin modulators might represent an option for resetting?

Along those lines, the authors report DOT1L and HDAC9 transcripts were down-regulated in the HM group. Is it possible to include chromatin details on this deregulation that the authors attribute to RAS pathway deregulation?

In addition, interdependence of BRD4 and DOT1L in leukemia has been reported (Gilan et al., 2016) suggesting that DOT1L, via dimethylated histone H3 K79, facilitates histone H4 acetylation, which in turn regulates the binding of BRD4 to chromatin. In the HM subgroup (which is also the one with poor prognosis) or in cell models how is the BRD4 expression and how is H3K79 methylation and H4 acetylation. May the DOT1L low expression contribute to chromatin closure influencing a reduction of acetylation and lower BRD4 binding at the interested areas?

In the literature a molecular link for RAS mediated transcriptional silencing and DNA hypermethylation is often reported (as also cited by the authors). The authors may like to comment in the discussion on the possibility of cross-talks and on a potential prioritization (if any) of genome/epigenome deregulation during leukemogenesis.

The authors state that data will be deposited in ERA, and accession codes will be provided during revision of the manuscript. This should be done.

Point-to-point reply (NCOMMS-17-09270-T)

Reviewers' comments:

Reviewer #1 Expert in leukaemia genetics:

The authors study the methylation patterns of DNA of patients with juvenile myelomonocytic leukemia (JMML). They hypothesized that DNA methylation profiling, alone or in combination with genetic alterations, could predict disease severity and outcome. DNA methylome analysis and mutation profiling was performed for 167 JMML samples. Clustering based on the variable CpG sites identified three clusters of patients (high, medium and low methylation), with the lowest methylation cluster having the highest rates of survival. The author thus demonstrated convincingly that methylation analysis has predictive value in JMML.

The authors look for the reason of the methylation changes, and propose a hypothesis, but fail to identify the cause of the methylation changes. The authors claim that differentially methylated sites were enriched for regions occupied by H3K27me3 or PRC2 and for genes associated with RAS signalling, but that part is relatively weak and only shown by indirect evidence.

Overall, the methylation data is nice and seems to predict clinical outcome very well. This is the strong part of the paper.

The reason why the DNA methylation is high or low in the different patients, is less clear, and based on data that is not very strong or not supported by data in JMML. It would be good to tune this down or include more data.

We thank Reviewer #1 for his overall positive feedback on our work. Following the reviewer's advise, we decided to carefully revise all sections of our manuscript dealing with the potential mechanism underlying the hypermethylation phenotype in order to make it clear that our findings suggest (but not prove) hyperactivation of the RAS-signaling pathway as the underlying molecular event in HM JMML cases based on correlative evidence.

Major remarks/questions:

1. Figure 1 demonstrates that a correction of the methylation data is required, for the various contributions of different cell types in each sample. Is this really needed? And how does the data look for JMML if no such correction is done. In the accompanying article (with some of the authors shared with this article), this seems not to be required.

We thank the reviewer for raising this important point. In fact, as also demonstrated in the accompanying manuscript by Stieglitz et al., cell type correction or exclusion of CpGs found to be dynamic during normal hematopoiesis is not essential for the detection of the JMML subgroups. We now included a heatmap depicting the consensus clustering results of JMML samples from the discovery cohort using the 5000 most variable CpGs without prior filtering (**Supplemental Figure 1A**). However, it is important to notice that 34.3% (1713) of the 5000 most variable CpG probes compared to only 16.1% (59230/367429) of all probes exhibited dynamic methylation in hematopoietic cells, demonstrating an enrichment of potentially confounding events in JMML. In addition, the decision to focus on CpGs that show no methylation dynamics in normal hematopoiesis was also based on recently published work done in our group showing that the epigenome of leukemic B-cells in chronic lymphocytic leukemia patients in large parts reflects the epigenome of the cell-of-origin. In this work, we were further able to demonstrate that epigenetic footprints pointing towards previously unknown pathogenetic events are unmasked only if physiologically occurring methylation programming is excluded from the analysis (Oakes et al., Nature Genetics 2016). In our present study of JMML, we were facing an even more complicated situation as only unsorted patient samples, containing varying proportions of different blood cells, were available for analysis (**Figure 1B**). In order to be able to identify JMML-specific methylation footprints, we decided to exclude all CpGs that exhibit dynamic methylation changes in normal hematopoiesis from all our analyses. In this way, our analysis excludes that the samples' cellular composition confound their cluster assignment, and, even more importantly, it prevents an erroneous confounding of hematopoietic differentiation patterns with pathogenetic events.

- Oakes CC, Seifert M, Assenov I, Gu L, Przekopowicz M, Ruppert AS, et al., DNA methylation dynamics during B cell maturation underlie a continuum of disease phenotypes in chronic lymphocytic leukemia. Nature Genetics, 2016. 48(3): p. 253-264.

2. Figure 5: methylation data is integrated with publicly available ChIP data for various histone marks (Encode data). This is of interest to generate a hypothesis, but would

need to be confirmed by ChIP experiments with JMML samples. In the current version of the manuscript, the conclusions are highly speculative and not confirmed by ChIP-seq data for JMML.

We agree with the reviewer that ChIP-seq data from JMML samples would be very valuable in order to draw specific conclusions. Unfortunately, for the vast majority of JMML samples, there is only DNA or frozen cell pellets available. Nevertheless, for the few samples for which we have access to viably frozen cells, we have already taken efforts in that direction: We have done Western blot and immunofluorescence analysis for several histone marks with the aim of performing ChIP-seq for those marks for which we see global differences, but unfortunately the unsorted, granulocyte-rich patient samples that were available, showed varying levels of protein degradation which made a robust interpretation of the data impossible. We therefore decided not to include any data from these experiments in the present study. We are now preparing to systematically collect sorted cell fractions from individual patients in a prospective manner in order to be able to perform more in depth molecular analyses in the future.

Similarly, figure 5H shows data on AML cell lines, which has little value. I would suggest to remove this part (figure 5) or to confirm the data in JMML samples with new experiments.

We thank the reviewer for this suggestion. In order to stay strictly in a human JMML disease context throughout the manuscript, we have now decided to take out the AML cell line methylomes figure panel (former **Figure 5H**) and to re-write the text accordingly to make it clear to the reader that we are talking about correlative evidence.

3. The DNA methylation profiles are clear, but have not been closely linked with expression data. Since DNA methylation is expected to influence expression, it would be a nice and essential addition to include a pairwise comparison between DNA methylation changes and RNA expression levels based on RNA-seq data. In the current manuscript this is only shown for a few selected genes.

&

4. Related to the previous remark: if RNA expression data is used for clustering, would this also result in a similar division over 3 clusters, with similar predictive value for survival/relapse?

We thank the reviewer for raising these important issues. We have now included a systematic comparison of DNA methylation and RNA expression. For 15/20 patients from the discovery cohort there was RNA of sufficient quality available to perform gene expression profiling using the Illumina HumanHT-12 v4 Expression BeadChips. The analysis of this data is summarized in **Supplemental Figure 6** of the revised version of our manuscript. Unsupervised clustering of the 1000 most variably expressed genes did not result in clusters that reflected survival/relapse of JMML patients, nor did the clustering recapitulate the methylation clusters (**Supplemental Figure 6A**). Nevertheless, globally, DNA methylation was inversely correlated with gene expression when considering all nCpGs located in gene promoters (**Supplemental Figure 6B**). We then correlated DNA methylation with gene expression across the JMML subgroups considering only the top 1000 genes with JMML-

specific methylation events (jmmIDMPs) in their promoters (**Supplemental Figure 6C**). As demonstrated before by our group and others, we found both positively and negatively correlated gene expression patterns, with a slightly higher proportion of negatively correlated genes (N=536 negatively correlated genes, N=464 positively correlated genes; Cabezas-Wallscheid et al., *Cell Stem Cell* 2014; Lipka et al., *Cell Cycle* 2014; Bock et al., *Molecular Cell* 2012). We then performed gene set enrichment analysis on this ranked gene list and found “Hallmark_KRAS_signaling_DN” to be significantly enriched in negatively correlated genes (**Supplemental Figure 6D**). Interestingly, we did not find any other enriched gene set, neither in the negatively nor in the positively correlated genes.

- Bock C, Beerman I, Lien WH, Smith ZD, Gu H, Boyle P, et al., *DNA methylation dynamics during in vivo differentiation of blood and skin stem cells*. *Mol Cell*, **2012**. 47(4): p. 633-47.
- Cabezas-Wallscheid N, Klimmeck D, Hansson J, Lipka DB, Reyes A, Wang Q, et al., *Identification of regulatory networks in HSCs and their immediate progeny via integrated proteome, transcriptome, and DNA Methylome analysis*. *Cell Stem Cell*, **2014**. 15(4): p. 507-22.
- Lipka DB, Wang Q, Cabezas-Wallscheid N, Klimmeck D, Weichenhan D, Herrmann C, et al., *Identification of DNA methylation changes at cis-regulatory elements during early steps of HSC differentiation using tagmentation-based whole genome bisulfite sequencing*. *Cell Cycle*, **2014**. 13(22): p. 3476-87.

Minor remarks:

5. Please explain abbreviations used, for example ‘DMPs’ is not explained as well as some other abbreviations.

We have now thoroughly revised the manuscript in order to ensure that all abbreviations used are explained in the text.

6. Figure 1B seems very black and white. There are no intermediate values. Is this correct?

Figure 1B, showing the cell type contribution for each sample as a heatmap does contain intermediated values. Did the reviewer maybe mean Figure 1D instead? -This panel displays the consensus clustering results and has mainly very high and very low values. This is due to the fact that the consensus clustering for 2 clusters was extremely stable as can also be discerned from the cluster consensus values which are 0.96 for the HM group and 1(!) for the LM group. This explains why the consensus matrix for k=2 indeed shows a “black & white” pattern. For three clusters (k=3), this looks different as can be seen in **Supplemental Figure 1F**.

Reviewer #2 Expert in leukaemia epigenetics:

In the present manuscript, the authors hypothesized that DNA methylation profiling, alone or in combination with genetic mutation profiling, might provide a molecular basis for JMML classification. With this approach the authors identified 3 JMML subgroups that showed unique molecular and clinical characteristics.

The study is clearly written and may lead to relevant prognostic data that might influence treatment of the JMML.

Here some suggestions to implement the preclinical and clinical value of the manuscript.

- 1. The authors indicate that up-regulation of DNMT1 and DNMT3B in the HM group is likely to be the molecular surrogate of the CIMP phenotype observed. The authors might include epigenome-based details on the cause of the identified up-regulation as well as they might verify using cell models if using chromatin modulators it is possible to re-equilibrate DNMTs expression of activity.**

We thank the reviewer for these suggestions. We have now included methylation profiles for candidate gene promoters in **Supplemental Figure 7**. We validated hypermethylation of the *AKAP12* promoter CpG island (**Supplemental Figure 7A**) and its correlation with down-regulation of *AKAP12* mRNA levels (**Figure 5F**). The promoter CpGs of *DNMT1*, *DNMT3A*, *DNMT3B*, and *TET2* showed low methylation levels around their transcription start sites (TSS), and methylation levels remained unchanged across all JMML methylation subgroups. In contrast, both *TET1* and *HDAC9* showed elevated methylation levels for promoter CpGs downstream of their TSSs, which would explain reduced mRNA expression levels. Indeed, we observed significant down-regulation of mRNA expression levels of *HDAC9* in primary JMML samples from the HM group as compared to the LM group (**Figure 5G**), whereas we could not detect a significant change in the mRNA expression levels of *TET1* in our patient cohort.

In summary, altered DNA methylation of their promoter regions does not explain the significant up-regulation of *DNMT1* and *DNMT3B* expression levels observed in our JMML cohort.

We agree with the reviewer that additional functional experiments would be needed in order to gain further mechanistic insight into the molecular mechanisms underlying the CIMP in JMML. Nevertheless, such studies are hampered by the lack of established cell line models for JMML. The use of mouse models could in principle substitute for studies in human cell lines, but such studies are very time-consuming and beyond the scope of the present study.

Another question that has also been raised by the reviewer was whether the CIMP in JMML could be modulated by the use of chemical compounds. Currently, such studies are hampered by the lack of appropriate disease models, but it is tempting to speculate that the recently described clinical response to DNMT-inhibitor therapy, which seems to work relatively well even in advanced and relapsed JMML cases (Furlan et al., Blood 2009; Locatelli and Niemeyer, Blood 2015), might be due to a reversal of the DNA hypermethylation present in JMML.

- Furlan I, Batz C, Flotho C, Mohr B, Lubbert M, Suttorp M, et al., *Intriguing response to azacitidine in a patient with juvenile myelomonocytic leukemia and monosomy 7*. *Blood*, **2009**. 113(12): p. 2867-8.
- Locatelli F, Niemeyer CM, *How I treat juvenile myelomonocytic leukemia*. *Blood*, **2015**. 125(7): p. 1083-90.

The authors also indicate that leukemias carrying RAS-pathway mutations exhibit methylation patterns akin to those observed in JMML (Fig. 5H). Could the authors use those cells to include some combo-treatments to verify whether targeting RAS (or the other) identified pathways together with potential chromatin modulators might represent an option for resetting?

We thank the reviewer for this suggestion. Based on the comments of reviewer #1, we have now removed Figure 5H from our manuscript in order to stay in the JMML context throughout the manuscript and not to dilute our findings with potentially confounding evidence derived from acute myeloid leukemia (AML) cell lines. Therefore, we have also decided not to follow the route of trying to perform pharmacologic epigenome modulation in AML cell lines.

Along those lines, the authors report DOT1L and HDAC9 transcripts were down-regulated in the HM group. Is it possible to include chromatin details on this deregulation that the authors attribute to RAS pathway deregulation?

We have now included the promoter methylation patterns of *DOT1L* and *HDAC9* in **Supplemental Figure 7E&H**. While the *DOT1L* promoter doesn't show any methylation differences across the methylation subgroups in JMML samples, the *HDAC9* promoter shows a significant increase in promoter methylation in both IM and HM JMML cases which is paralleled by a significant downregulation of *HDAC9* mRNA expression levels. Indeed, it would be interesting to understand which other chromatin factors (if any) are changing across JMML subgroups, but unfortunately, primary patient material of sufficient quality that would be amenable to ChIP(-seq) experiments is not available. We are currently working to establish a prospective collection of viably frozen cells from JMML patients in order to facilitate such analyses for future studies.

In addition, interdependence of BRD4 and DOT1L in leukemia has been reported (Gilan et al., 2016) suggesting that DOT1L, via dimethylated histone H3 K79, facilitates histone H4 acetylation, which in turn regulates the binding of BRD4 to chromatin. In the HM subgroup (which is also the one with poor prognosis) or in cell models how is the BRD4 expression and how is H3K79 methylation and H4 acetylation. May the DOT1L low expression contribute to chromatin closure influencing a reduction of acetylation and lower BRD4 binding at the interested areas?

We thank the reviewer for suggesting this hypothesis. Although we do not have ChIP-sequencing data on primary JMML samples for the reasons detailed above, we have carefully screened public databases for existing ChIP-sequencing data sets that would help us to address this hypothesis. As depicted in **Supplemental Figure 5C**, H3K79me2 ChIP-seq peaks (among other activating histone marks) are significantly under-represented in

jmmIDMPs while H4k20me1 shows slight and H3K9me3 & H3K27me3 show strong enrichment in jmmIDMPs. In addition, we have investigated a published BRD4 ChIP-seq data set and found a strong depletion of BRD4 binding sites in the jmmIDMPs (please refer to the figure below). Together these data indicate that jmmIDMPs are neither decorated with H3K79me2 nor bound by BRD4 under normal conditions. Nevertheless, we consider it worthwhile testing this potential interplay between DOT1L and BRD4 in primary cells or in a murine disease model using ChIP-seq in future studies.

Enrichment analysis of transcription factor ChIP-seq peaks. ChIP-seq peak files were downloaded from the CODEX website (<http://codex.stemcells.cam.ac.uk/>). All jmmIDMPs were used as foreground, and the remaining non-differentially methylated nvCpG sites form a background set. Enrichment analysis was performed independently for every pair (state/cell line) on the foreground set as compared to the background set. Logarithmic fold change values were calculated as $\log_2(OF/OB)$, where OF is the fraction of probes in the foreground set overlapping with the chromatin state type of interest, and OB is the same metric for the background set. P-values were obtained using Fisher's exact test and adjusted for multiple testing using the Benjamini-Hochberg method. The significance threshold applied was 0.01.

In the literature a molecular link for RAS mediated transcriptional silencing and DNA hypermethylation is often reported (as also cited by the authors). The authors may like to comment in the discussion on the possibility of cross-talks and on a potential prioritization (if any) of genome/epigenome deregulation during leukemogenesis.

We thank the reviewer for this suggestion. We have now included this point in the discussion section as follows (page 15, lines 391-397):

“One might speculate that these secondary mutations contribute to transcriptional deregulation of DNMT1 and thus further augment the extent of epigenetic remodeling. Alternatively, pre-existing epigenetic alterations might provide a fertile ground for malignant transformation following single or few genetic hits. This sequence of events has been shown recently in the context of cigarette smoke-induced lung cancer where DNA hypermethylation of PRC2 target genes sensitizes bronchial epithelial cells to single-step transformation by mutant KRAS.”

The authors state that data will be deposited in ERA, and accession codes will be provided during revision of the manuscript. This should be done.

The Illumina Infinium HumanMethylation450 Bead Chip Array data as well as the exome-seq data have been deposited in ERA (<https://www.ebi.ac.uk/ega/home>) under study number

EGAS00001002511. The submission of the gene expression array data is currently under way and this data set also will be deposited under the same study number.

REVIEWERS' COMMENTS:

Reviewer #1 (Remarks to the Author):

The authors have submitted a very nice revised manuscript, with attention for all comments. I have no further remarks.

Reviewer #2 (Remarks to the Author):

The authors have added data in support of the implementation of the manuscript.

1-In relation to the data provided showing that altered DNA methylation of their promoter regions does not explain the significant up-regulation of DNMT1 and DNMT3B expression levels observed in our JMML cohort, the authors should highlight hypothesis in the discussion on the potential mechanistics.

2. A comment on the potential hypothesis of the interplay between DOT1L and BRD4 might be added in the discussion

Point-to-point reply (NCOMMS-17-09270A)

Reviewer #1 (Remarks to the Author):

The authors have submitted a very nice revised manuscript, with attention for all comments. I have no further remarks.

Reviewer #2 (Remarks to the Author):

The authors have added data in support of the implementation of the manuscript.

- 1. In relation to the data provided showing that altered DNA methylation of their promoter regions does not explain the significant up-regulation of DNMT1 and DNMT3B expression levels observed in our JMML cohort, the authors should highlight hypothesis in the discussion on the potential mechanistics.**

We thank the reviewer for this suggestion. We have now added a sentence to the results section (line 475) that discusses the potential reasons for a lack of detection of aberrant DNA methylation in the promoters of DNMT1 and DNMT3B in HM JMML samples.

- 2. A comment on the potential hypothesis of the interplay between DOT1L and BRD4 might be added in the discussion**

This is an interesting hypothesis for which we tried to find evidence in our study during the first revision. However, we were not able to find support for this hypothesis in our data sets. Therefore, we believe that this hypothesis warrants further investigation in future studies using dedicated experiments.